# Developing the Techniques for Solving the Inverse Problem in Photoacoustics

**Mioljub Nesic \*** 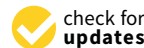**, Marica Popovic and Slobodanka Galovic**

Department of Atomic Physics, Vinca Institute for Nuclear Sciences, 11000 Belgrade, Serbia;
maricap@vin.bg.ac.rs (M.P.); bobagal@vin.bg.ac.rs (S.G.)
**\*** Correspondence: mioljub.nesic@vin.bg.ac.rs; Tel.: +381-64-1417-188

**Abstract:** In this work, theoretically/mathematically simulated models are derived for the photoacoustic (PA) frequency response of both volume and surface optically-absorbing samples in a minimum volume PA cell. In the derivation process, the thermal memory influence of both the sample and the air of the gas column are accounted for, as well as the influence of the measurement chain. Within the analysis of the TMS model, the influence of optical, thermal, and elastic properties of the sample was investigated. This analysis revealed that some of the processes, characterized by certain sample properties, exert their dominance only in limited modulation frequency ranges, which are shown to be dependent upon the choice of the sample material and its thickness. Based on the described analysis, two methods are developed for TMS model parameter determination, i.e., sample properties which dominantly influence the PA response in the measurement range: a self-consistent procedure for solving the exponential problems of mathematical physics, and a well-trained three-layer perceptron with back propagation, based upon theory of neural networks. The results of the application of both inverse problem solving methods are compared and discussed. The first method is shown to have the advantage in the number of properties which are determined, while the second one is advantageous in gaining high accuracy in the determination of thermal diffusivity, explicitly. Finally, the execution of inverse PA problem is implemented on experimental measurements performed on macromolecule samples, the results are discussed, and the most important conclusions are derived and presented.

**Keywords:** photoacoustic; photothermal; inverse problem; thermal memory; minimum volume cell; neural networks; thermal diffusivity; conductivity; linear coefficient of thermal extension

## 1. Introduction

One of the most plastic and easily understandable definitions of inverse problem was given by professor Mandelis [1]—a field in which one is called upon to reconstruct the cow from the hamburger meat. Indeed, when all the difficulties are taken into considerations, such as ill conditioning, non-linearity, model dependence upon material, experimental range limitations, etc., one truly feels like they are dealing with the impossible. On the other hand, no matter what method of inverse problem solving is opted for, one conclusion seems inevitable—it is necessary to simultaneously develop both the appropriate TMS model (direct solving methods) and the inverse solving procedures (characterization, imaging) in order to obtain optimum results. By reviewing literature regarding TMS models and techniques of inverse solving in photoacoustics, unexplained approximations that could be the limiting factor in determination of sample properties were noticed.

The research in this domain done by our group has taken two directions. From the experimental point of view, it was found that only a narrow bandwidth of frequency measurements has been exploited until now; from our own experience, this is due to the fact that experiential results rarely

agree with theoretical predictions over the entire frequency range. Also, the processing of results, almost by rule, considers either amplitude or phase measurements of the signal; never does it account for both of those, simultaneously.

From the theoretical modeling aspect (the aspect of fundamental research), it was found that the influence of finite heat propagation velocity was neglected, as well as the influence of volumetric optical absorption and the possibility of multiple optical reflections. Also, the knowledge of the measurement chain influence can be, in general, considered insufficient, and it plays an important role in the process of obtaining experimental results.

That is why, in the first part of this work, the generalized model of photoacoustic (PA) response was presented and discussed as the basis for the developed inverse solving procedures for PA characterization. Furthermore, two types of inverse problem solving are suggested and analyzed: a self-consistent inverse procedure, and a neural network. Finally the results of the application on experimental results of the first method are presented. At the end, the most important conclusions are derived.

## 2. Generalized Model of PA Response—Direct Problem Solving

Indirect transmission photoacoustics presumes the use of an air-filled PA cell as the element in which the acoustic signal is created due to the deployment of a monochromatic, amplitude modulated light source: $I = I_0(1 + \cos \omega t)$ upon a sample. Usually, a cylindrical cell is used in combination with a disk-shaped sample of the radius $R$ and the thickness $l_s$, positioned and fixed in accordance to the "simply supported plate" principle [2]. This sample is exposed, from one side, to the described EM source, while the response is recorded by microphone on the other side, i.e., this is the principle of t transmission gas-microphone configuration, presented in Figure 1a (upper part), while a detailed description is given in Figure 2. The frequency-dependent measurements of PA response are performed using a lock-in amplifier.

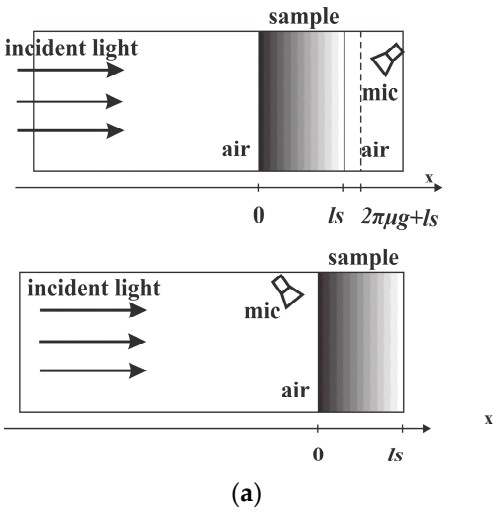
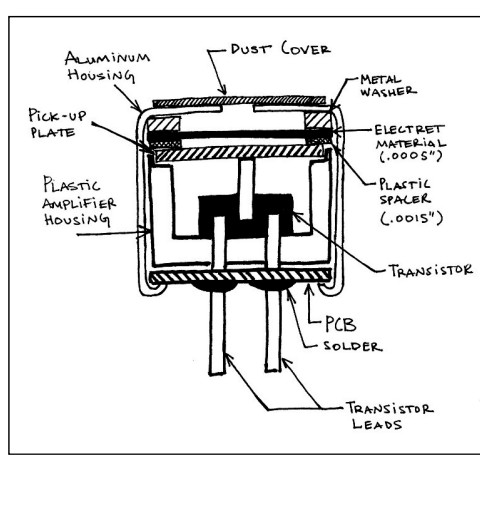

**(a)**　　　　　　　　　　　　　　　　　　　　　　　**(b)**

**Figure 1.** (**a**) Schematic representation of transmission (up) and reflection (down) configuration in photoacoustics, (**b**) drawing of a standard electrets microphone (http://www.openmusiclabs.com/wp/wp-content/uploads/2011/03/mic_section_small.jpg, accessed on 29 November 2018).

In transmission PA configuration Figure 1a, the concept of minimum volume cell is used in order to obtain sufficiently high-measured acoustic signal and good signal-to-noise ratio. This means that the microphone chamber itself acts as the interior of the PA cell, as illustrated in Figure 1b [3,4].

Based upon previous research and in accordance with literature defined norms [5], the following designation of thermodynamic properties of the system is introduced:

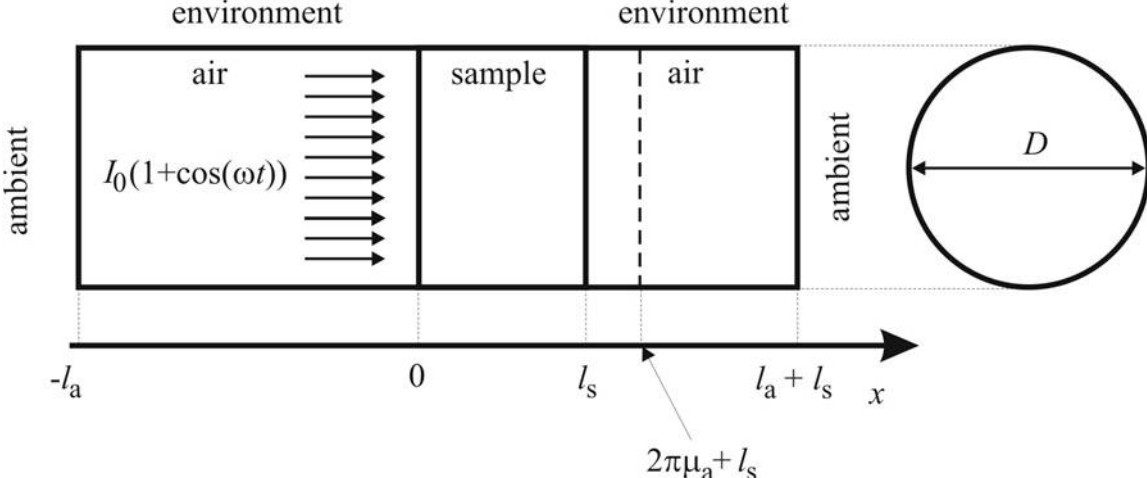

**Figure 2.** Transmission configuration, taken with approval from [6].

$k_i$—*thermal conductivity* [W/mK],
$D_{Ti}$—*thermal diffusivity* [m²/s],
$a_T$—*thermal coefficient of linear expansion* [K⁻¹],
$\tau_i$—*thermal relaxation time* [s],
and $u_i = \sqrt{D_{Ti}/\tau_i}$—*heat propagation velocity* [m/s], indexed $i = a,s$ (*air or sample*) and designating the *i*-th medium where it occurs.

Thermal relaxation time and heat propagation velocity are the properties of materials which exist in the generalized theory of heat transfer [6–12]. Explanation attempts regarding the meaning of thermal relaxation time can be found in several papers [13–15]. Avoiding further considerations of the matter (since they go beyond the scope of this work), it is important to note that the investigations in this area are still ongoing and can be approached from different physical viewpoints. As for the stand of our group, we have adopted the most general interpretation of this property, regardless from the material microstructure or thermal energy carriers—thermal relaxation time is the period of time passing between the occurrence of the excitation and the actual change of the heat flux.

Prior to the development of the theoretical/mathematical simulation (TMS) model of the generalized PA response, the following presumptions were introduced:

(a) The cross section of the incident beam is much larger than thesurface area of the sample, thus, planar uniformity of energy distribution justifies the use of one-dimensional (*1D*) approximation [4,8,16–21];

(b) Excitation energy is absorbed within thin surface layer of the sample (the approximation which describes metal samples well, otherwise achieved through the application of thin, opaque absorbent layer) [18,20];

(c) Heat conduction to the surrounding gas (outside of the PA cell) is considered negligible due to its poor thermal conduction properties [18];

(d) Harmonic component alone (of the Fourier transform of the acquired signal) is observed (*lock-in* detection) and frequency characteristics of the PA response are analyzed;

(e) The "thin plate" approximation is applicable since $R$ is much larger than $l_s$, where the influence of the sample dilatation on the mechanical piston model is negligible and only *thermoelastic* (TE) bending is taken into consideration [4,8,18,19,22,23].

Basing the approach upon literature considerations [4,8,18,19,22,23], the starting expression of the model (for the measured signal directly proportional to the pressure change in the PA cell) can be written in the form:

$$\widetilde{p} = \widetilde{p}_{th} + \widetilde{p}_{ac} \tag{1}$$

where $p_{th}$ denotes the pressure change due to the thermoconducting(*TH*) component of the PA response (the component that originates from the periodic expansion of a thin gas layer closest to the sample in the PA cell), while $p_{ac}$ denotes the pressure change due to the PA component originating from TE vibrations of the sample caused by temperature gradient along symmetry axes of sample (drum effect). These components are then written in the following manner [4], [18], [8]:

$$\widetilde{p}_{th} = \frac{\gamma P_0}{l_a T_0} \int\limits_{l_s}^{l_s + 2\pi\mu_a} \widetilde{\vartheta}_s(l_s) e^{-\widetilde{\sigma}_a(x - l_s)} dx, \tag{2a}$$

$$\widetilde{p}_{ac} = \frac{3\gamma P_0}{l_a} \alpha_S \frac{R^2}{l_s^3} \int\limits_0^{l_s} (x - \frac{l_s}{2}) \widetilde{\vartheta}_s(x) dx, \tag{2b}$$

where $\gamma$ annotates the adiabatic coefficient, $P_0$ is the atmospheric pressure, $l_a$ is the length of the gas column inside the PA chamber, while $T_0$ stands for the room temperature. Furthermore, $\alpha_S$ is the linear coefficient of thermal expansion of the sample, $R$ is its radius, while $l_s$ annotates its thickness. The symbols $\widetilde{\vartheta}_s$ and $\mu_a$ represent the complex representative of distribution of dynamic temperature variations across the sample and the thermal diffusion length in the air (gas).

As can be seen from the expressions (1) and (2), the components of the pressure depend on the distribution of the dynamic temperature variation along the sample axis and from the dynamic temperature variation on the unexposed side of the sample (back). We have acquired these values taking into consideration finite heat propagation velocity [9]:

$$\frac{d^2 \widetilde{\vartheta}_a(x, \omega)}{dx^2} - \widetilde{\sigma}_a^2 \widetilde{\vartheta}(x, \omega) = 0, \tag{3a}$$

$$\frac{d^2 \widetilde{\vartheta}_s(x, \omega)}{dx^2} - \widetilde{\sigma}_s^2 \widetilde{\vartheta}_s(x, \omega) = -\widetilde{\sigma}_s \widetilde{z}_{cs} S(x) \tag{3b}$$

$$\widetilde{q}_i(x, \omega) = -\frac{1}{\widetilde{\sigma}_i \widetilde{z}_{ci}} \cdot \frac{d\widetilde{\vartheta}_i(x, \omega)}{dx}, \quad i = a, s \tag{3c}$$

where q(x) is dynamic heat flux, $S(x)$ represents incident volumetric heat flux which generates perturbations of temperature field, $\widetilde{\sigma}_i$ and $\widetilde{Z}_{ci}$, are *heat wave vector* and *thermal impedance* of the environment (air or sample), given by:

$$\widetilde{\sigma}_i = \frac{1}{\sqrt{D_{Ti}}} \sqrt{j\omega(1 + j\omega\tau_i)}, \tag{4a}$$

$$\widetilde{Z}_{ci} = \frac{\sqrt{D_{Ti}}}{k_i} \sqrt{\frac{(1 + j\omega\tau_i)}{j\omega}}, \tag{4b}$$

Incident volumetric heat flux is calculated as:

$$S(x) = -\frac{dI_{abs}(x)}{dx} \tag{5a}$$

$$I_{abs}(x) = I_0 e^{-\beta x} \tag{5b}$$

$$I_{abs}(x) = I_0(1 - R_0)\frac{e^{-\beta x}}{1 + Re^{-\beta x}} \tag{5c}$$

The equations for the components of the measured pressure (2a,b), combined with the expressions (3a)–(5c) and solved, become:

$$\widetilde{p}_{th} = \frac{\gamma P_0}{l_a T_0} \frac{1}{\widetilde{\sigma}_a} e^{2\pi\mu_a} \vartheta_s(l_s) \tag{6a}$$

$$\widetilde{\vartheta}_S(l_s) = -\frac{S_0\beta\widetilde{\sigma}_s\widetilde{Z}_{cs}}{\beta^2-\widetilde{\sigma}_s^2} \cdot \frac{e^{(\sigma_a-\beta)l_s}\left[(\widetilde{r}+\widetilde{r}_a)ch(\widetilde{\sigma}_sl_s)+(1+\widetilde{r}_a^2)sh(\widetilde{\sigma}_sl_s)\right]+(\widetilde{r}-\widetilde{r}_a)e^{-\sigma_sl_s}}{2\widetilde{r}_ach(\widetilde{\sigma}_sl_s)+(1+\widetilde{r}_a^2)sh(\widetilde{\sigma}_sl_s)} \cdot \frac{1-R_0}{1+R_1e^{-\beta l_s}},$$
$$\left(\widetilde{r}=\frac{\beta}{\widetilde{Z}_{cs}}, \widetilde{r}_a=\frac{\widetilde{Z}_{ca}}{\widetilde{Z}_{cs}}\right) \tag{6b}$$

$$\widetilde{p}_{ac} = S_0\frac{6\gamma P_0 R^4}{l_a l_s^2 R_c^2}\alpha_s\frac{\widetilde{z}_{cs}}{\widetilde{\sigma}_s^2}\frac{ch(\widetilde{\sigma}_sl_s)-\frac{\widetilde{\sigma}_sl_s}{2}sh(\widetilde{\sigma}_sl_s)-1}{sh(\widetilde{\sigma}_sl_s)}\frac{1-R_0}{1+R_1e^{-\beta l}} \tag{6c}$$

where $R_0$ and $R_1$ are, respectively, outer and inner optical reflection coefficient.

If the sample is good optical absorber, heat source becomes surface type and the model given by (6a–c) is reduced to:

$$\widetilde{p}_{th} = S_0\frac{\gamma P_0}{T_0 l_a}\frac{\widetilde{Z}_{cs}}{\widetilde{\sigma}_a}\frac{1}{sh(\widetilde{\sigma}_sl_s)}, \tag{7a}$$

$$\widetilde{p}_{ac} = S_0\frac{6\gamma P_0 R^4}{l_a l_s^3 R_c^2}\alpha_s\frac{\widetilde{Z}_{cs}}{(\widetilde{\sigma}_s)^2}\frac{ch(\widetilde{\sigma}_sl_s)-\frac{\widetilde{\sigma}_sl_s}{2}sh(\widetilde{\sigma}_sl_s)-1}{sh(\widetilde{\sigma}_sl_s)}. \tag{7b}$$

In the above expressions $S_0$ stands for the surface heat source, which equals half of the excitation energy intensity, $R_c$ represents the effective radius of the sample [24], and $\omega = 2\pi f$ is radial modulation frequency.

Finally, the PA response is given in the form:

$$\widetilde{p}_{ins} = E_0\frac{1}{1+j\omega\tau_e}(\widetilde{p}_{th}+\widetilde{p}_{ac}) = E_0\frac{\widetilde{p}_{th}}{1+j\omega\tau_e}\left(1+\frac{\widetilde{p}_{ac}}{\widetilde{p}_{th}}\right). \tag{8}$$

In the expression (8), the influence of the measurement chain is represented through the presence of the element $E_0/(1+j\omega\tau_e)$, which can, however, be annulated by diverse normalization procedures [20,23].

When the influence of thermal memory is neglected, the expressions (5a,b) and (7) are reduced to their classic *composite piston* forms [2,4,8,18,19,22,23].

## 2.1. Multiple Optical Reflections—the Influence of Optical Properties

In thin samples with low optical absorption coefficient an increase of the static component of the PA response temperature variation was calculated as:

$$\Theta_{(0)}^{(s)} = \Theta'^{(s)}_{(0)} \cdot \frac{1}{1+Re^{-\beta l_s}}. \tag{9}$$

On the other hand, in thin samples ($l_s$~10 μm, optical absorption coefficient $\beta$ ~$10^5$ m$^{-1}$, typical for polymers) with high inner reflection coefficients ($R_1$~0.9), at low frequencies (100 Hz–10 kHz), temperature variation (the dynamic component of the signal) is significantly increased, more noticeably on the exposed side of the sample ($x = 0$), as presented in Figure 3a.

However, in thick samples (~100 μm), as well as in others with low inner reflection coefficients ($R_1$~0.1), the effect is, surprisingly, the opposite: the temperature variation is decreased compared to the one corresponding to the model which neglects multiple reflections—the effect which can be seen in Figures 3b and 4a,b. Graphic representation of this principle is given in Figure 5.

These considerations present us with the possibility of observing another TMS model parameter—optical coefficient of inner reflection—in the future, but also call for caution; fundamental aspects of heat transfer through various media should be more profoundly studied [25].

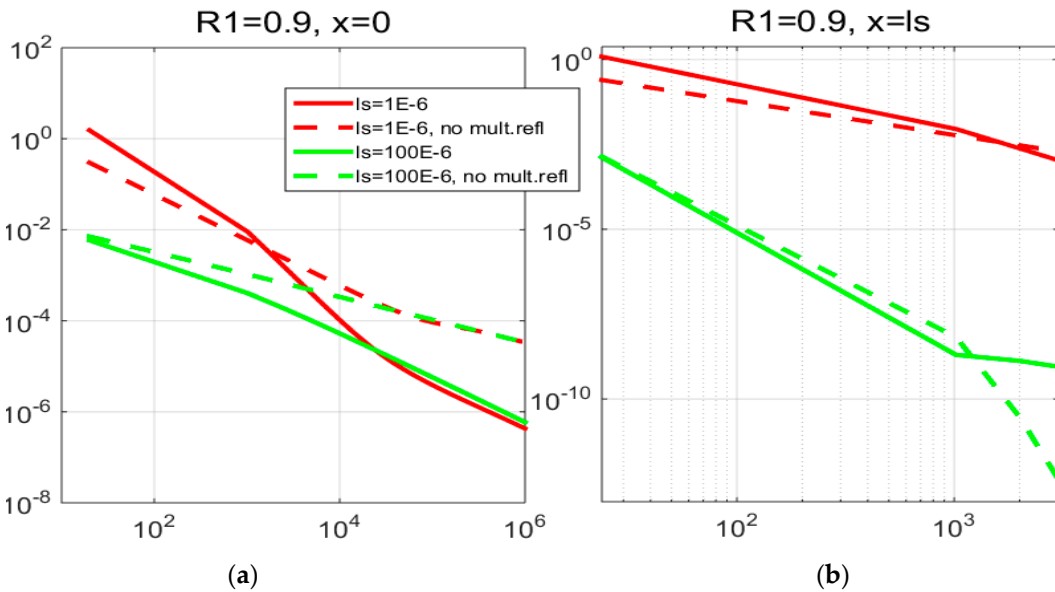

**Figure 3.** The influence of multiple optical reflections on surface temperature variation of samples with high inner reflection coefficient.

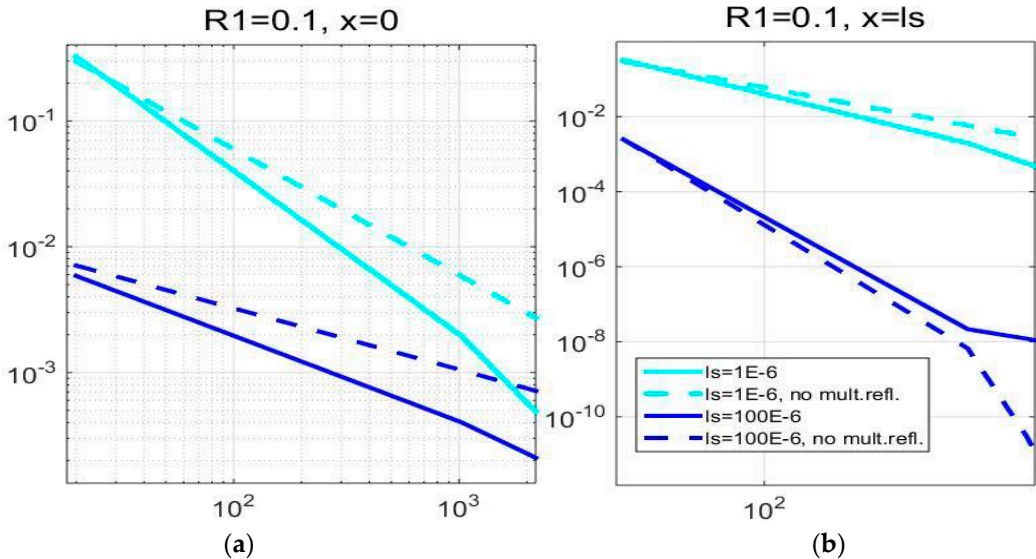

**Figure 4.** The influence of multiple optical reflections on surface temperature variation of samples with low inner reflection coefficient.

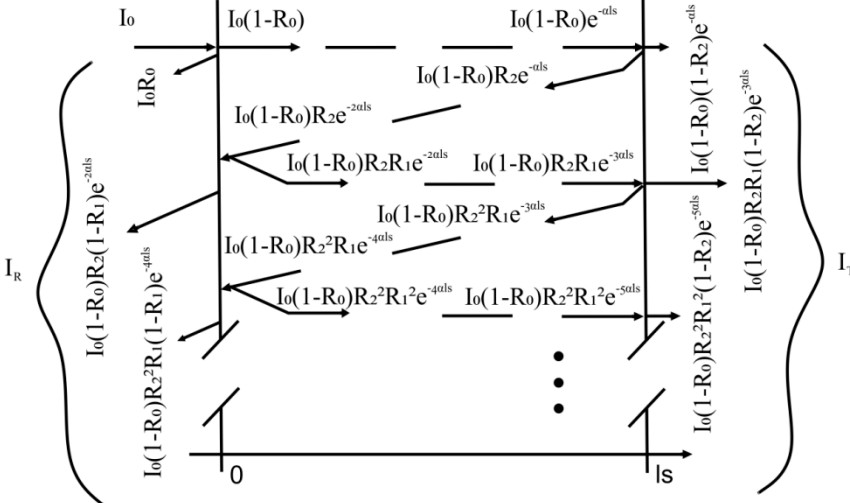

**Figure 5.** Graphic representation of the principle of multiple optical reflections.

## 2.2. Thermal Memory Influence

In numerous materials and under certain conditions, the appearance of oscillatory behavior as well as shape changes in both phase and amplitude responses are predicted; however, due to technical limitations of the experiment, these could not be recorded and validated. Instead, theoretical predictions are presented for reference samples (Aluminum, two thickness levels) in Figure 6.

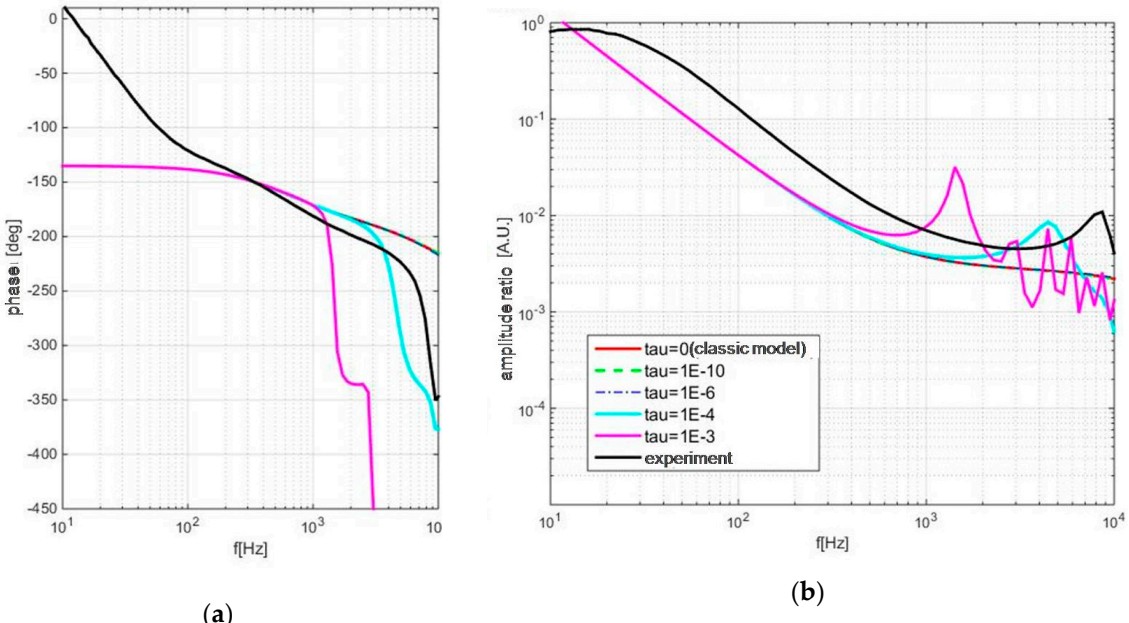

(**a**)

(**b**)

**Figure 6.** *Cont.*

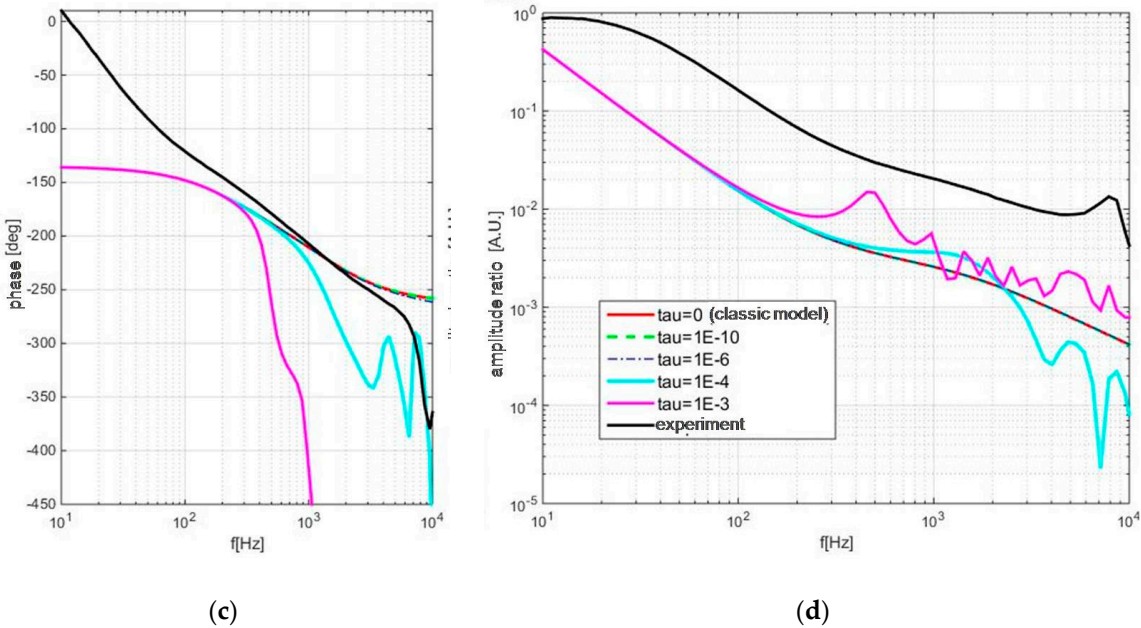

(**c**)  (**d**)

**Figure 6.** Thermal memory influence on PA response of aluminum samples 100 (**a**,**b**) and 300 (**c**,**d**) microns thick: phase is given on the left (**a**,**c**) and amplitude is on the right (**b**,**d**).

The most important result of these considerations is the expression which directly links the position of the first peak (the frequency value) with heat propagation velocity through the observed medium [9,10,12,21,26]:

$$f_{1\max} = \frac{1}{2l_s}\sqrt{\frac{D_{Ts}}{\tau_s}} = \frac{1}{2l_s}u_s. \tag{10}$$

### 2.3. Helmholtz Resonances—the Influence of the Measurement Chain

Resonant peaks observed in this part of frequency domain are, throughout literature, attributed to the influence of measurement chain, although, in measurements, they occur at frequencies lower than expected (frequency characteristic of the microphone, the amplifier, and other electronics) [27–29]. Minimum volume cell has already been observed as an electro-acoustic resonator and it has been modeled with cascade filter array, with transfer function represented as the combination of two Helmholtz resonators:

$$\widetilde{p}_u(j\omega) = \widetilde{p}(j\omega) \cdot H_V(j\omega) \cdot H_\varepsilon(j\omega). \tag{11}$$

The relation among different elements of the analogous electro-acoustical system and the actual geometrical values of the microphone are given in the following set of expressions [30]:

$$L = \frac{\rho l}{S}, C_i = \frac{V_i}{\rho v^2}, i = V, \varepsilon, \omega_{closed} = v\sqrt{\frac{S}{lV}},$$

$$H_i(j\omega) = \frac{\omega_i^2}{\omega_i^2 - \omega^2 + j\omega\frac{\omega_i}{Q_i}}, \quad \begin{bmatrix} i = V, \varepsilon \\ \omega = 2\pi f \\ \omega_i = 2\pi f_i \end{bmatrix}, \tag{12}$$

while the graphical representation of the analogy is given in Figure 7:

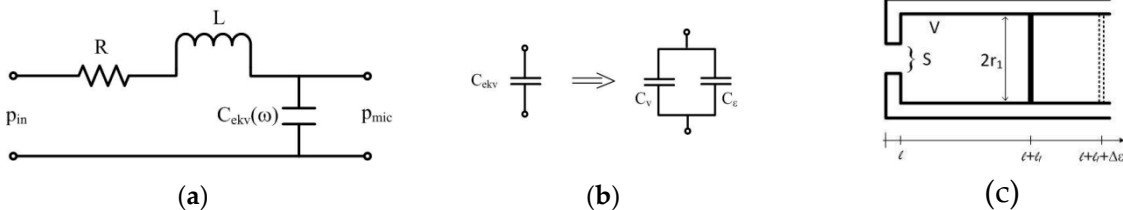

**Figure 7.** (**a**) Analogous electrical circuit, (**b**) capacitance model of the microphone chamber, (**c**) actual geometry of the microphone chamber.

The results of the application of the model are given in Figure 8:

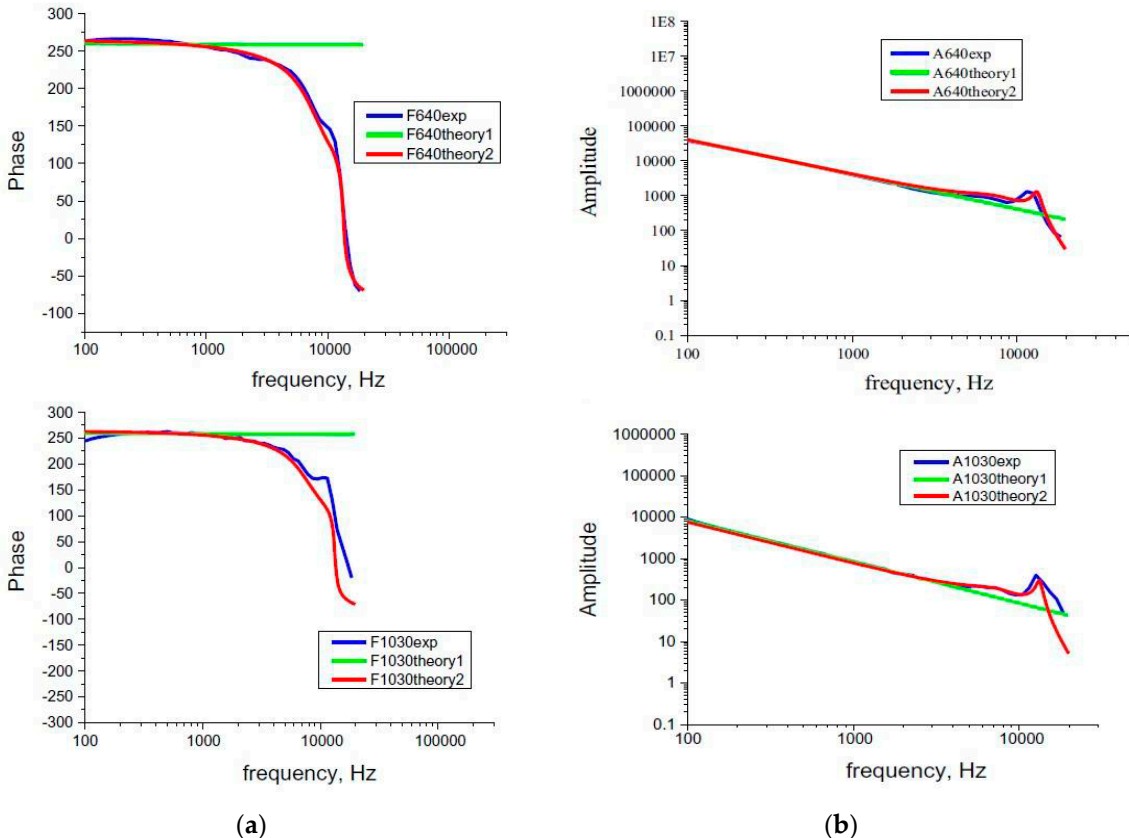

**Figure 8.** The inclusion of Helmholtz resonances in a PA experiment post-processing (**red line**): phase (**a**) and amplitude response (**b**), theoretical prediction (**green**), experimental results (**blue**) of HDPE at 640 and 1000 microns.

These considerations not only presented us with the possibility of effectively eliminating microphone influence in our future experiments, but also open the possibility of introducing a novel method of microphone characterization.

## 3. Techniques for Inverse Solving of PA Response

Based on the described analysis, two methods are developed for TMS model parameter determination, which were applied on numerical experiments:

1. A self-consistent procedure for solving the exponential problems of mathematical physics;
2. A well-trained three-layer perceptron with back propagation, based upon theory of neural networks.

　　　The first method was, consequentially, applied on experimental measurements, with satisfactory results, and published [31] (subsection 2.2.3).

### 3.1. Self-Consistent Inverse PA Procedure

　　　The idea for the development of the self-consistent inverse procedure for the estimation of thermodynamic parameters originates from theoretical considerations of PA model, where the tendency of phase exhibiting linear dependence upon thermal diffusivity, $D_{Ts}$, was noticed. The benefit of this approach is that this parameter, when derived from phase data, improves the reliability of multi-parameter fitting done on the rest of the signal (amplitude data). As a matter of fact, analytical methods demonstrated that thermal conductivity, $k_s$, could not be identified separately, but only as the part of its ratio with linear expansion coefficient, $\alpha_s$: $\frac{\alpha_T}{k_s}$ [32], which boils the fitting procedure down to only one parameter.

　　　The validity of the idea was demonstrated first by TMS modeling of the problem, i.e., on a numerical experiment, presented in Figures 9 and 10. In Matlab package, the procedure was developed which randomly sets the values of the dataset $D_{Ts}$, $\frac{\alpha_T}{k_s}$ (in accordance to literature values), and then simulates the PA response at two thickness levels using the given set of parameters with the addition of the certain level of noise. In the next step, the estimation of $D_{Ts}$ is done by the comparison of phase difference data, while the value of $\frac{\alpha_T}{k_s}$ is estimated from the amplitude ratio—both estimates are done by regression analysis: least squares being the method of choice.

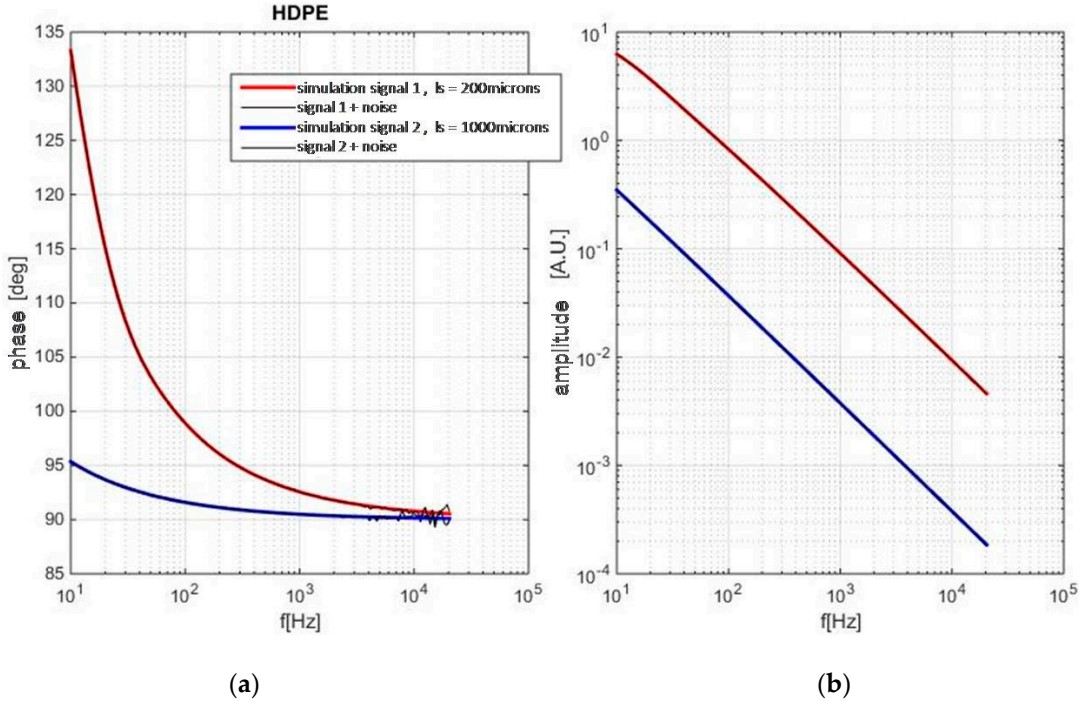

**Figure 9.** Simulated PA response at two thickness levels, with the addition of noise: phase (**a**) and amplitude (**b**).

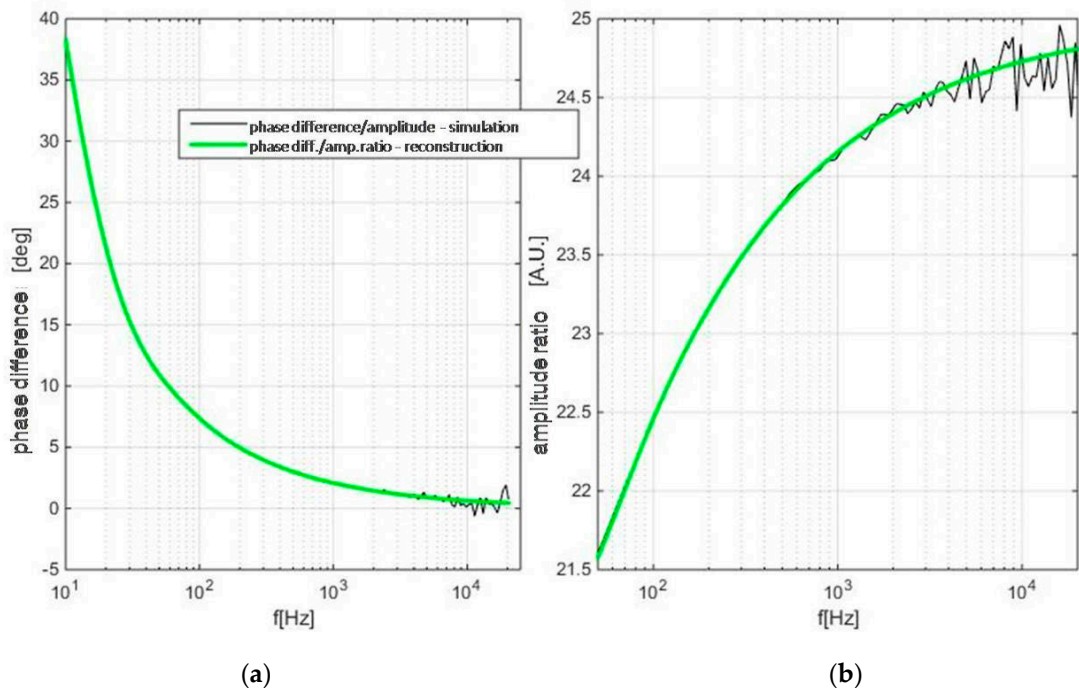

**Figure 10.** Simulated (**black line**) and reconstructed PA response (**green**), based upon the estimated values of parameters.

The results were quite interesting: after 1000 iterations, the procedure retuned the error for $D_{Ts}$ 0.94% (always under 2%) and 44.15% for $\frac{\alpha_T}{k_s}$ (always above 30 %!). The conclusion was drawn that some parts of the model must be seriously ill-conditioned in the case of soft matter materials.

*3.2. The Application of the Neural Network*

Finally, a neural network was developed in order to assess the ill-conditioning issue of the inverse problem in photoacoustics of polymers. The type was multilayer perceptron, learning method was back-propagation, and the input parameters: $k_s, D_{Ti}, \alpha_T, l_s$. The material of choice: HDPE. Training was done on 40% of the sample dataset, 10% was used for validation, and testing (reconstruction) was applied on 50% of it. The results, after 10000 simulations were more than satisfying: estimation error for $\frac{\alpha_T}{k_s}$ was as low as 0.71%! As for the accuracy was not uniform: for low and high values of the parameter, the error was noticeable, but still, for the most of the sample set it remained under 2.15%!

However, what was more important than the estimation results, themselves, was the accompanying analytics, which, for the first time, presented the graphical representation of the ill-conditioning of the model, itself! Figure 11 clearly indicates how steep the dependence upon two parameters can be in the case of $D_{Ts}$.

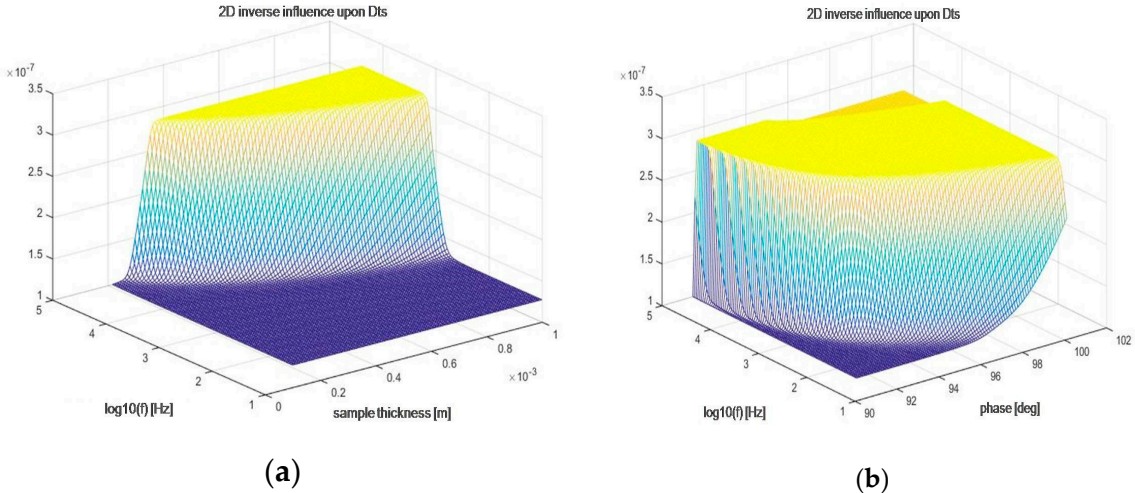

(**a**)                                                    (**b**)

**Figure 11.** The dependence of $D_{Ts}$ upon two parameters: thickness and modulation frequency (**a**) and phase and modulation frequency (**b**).

Finally, a conclusion could be drawn that, in case of soft matter materials such as HDPE, a self-consistent procedure could be more adequate for the estimation of $D_{Ti}$, while neural network approach clearly stood out when the estimation of $\frac{\alpha_T}{k_s}$ is concerned.

### 3.3. The Application on Experimental Data

The pioneering paper concerning the application of self-consistent procedure for the estimation of thermodynamic parameters on experimental data was published in 2018 [31]. HDPE samples had been prepared and characterized in advance at "Vinca" Institute for Nuclear Sciences in such a manner that their thickness and chemical or structural composition could not be questioned [31,33–36]. Using methods such as wide angle X-ray diffraction (WAXD) and diffraction scanning calorimetry (DSC), it was proven that regular normalization method (on two levels of thickness) could not be deployed. Also, crystallinity levels were estimated and are presented in Table 1.

**Table 1.** Crystallinity—functional dependence upon preparation conditions and sample thickness.

| $\chi$ (%) | 200 μm | | 400 μm | 600 μm |
|---|---|---|---|---|
| | *DSC* | *WAXD* | *DSC* | *DSC* |
| *Fast Cooled* | 51.7 | 50.5 | 57.4 | 59.3 |
| *Slowly Cooled* | 73.8 | 72.5 | 71.5 | 70.8 |

Regression analysis of the difference between theoretical prediction and the experimentally obtained PA response demonstrated that thin samples (200μm) have the potential for differentiating between different levels of crystallinity, as presented in Figure 12.

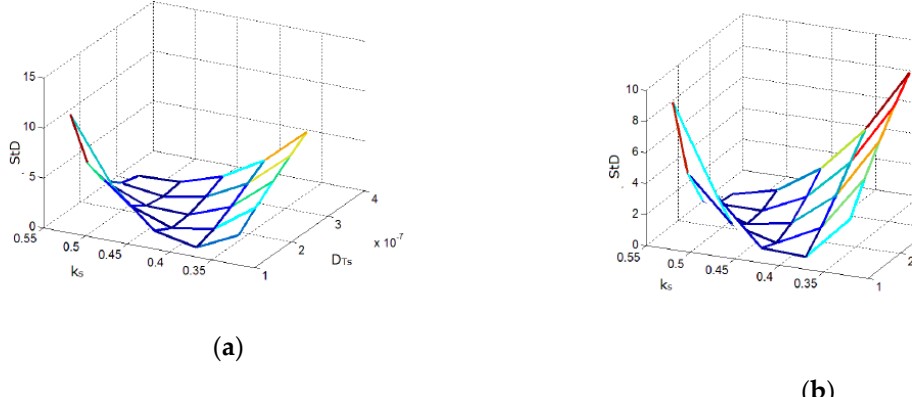

**Figure 12.** Deviation over region of interest as function of (ks, DTs) datasets for 200μm thick samples at two levels of crystallinity: ~70% (**a**) and ~50% (**b**).

Estimated results are presented in Table 2.

**Table 2.** Estimated values of thermodynamic parameters of HDPE (uncertainty given as half-distance between points).

| Thickness [μm] | | HDPE—High-Density Polyethylene | | |
| --- | --- | --- | --- | --- |
| | | *Fast Cooled* | *Slowly Cooled* | *Uncertainty* |
| 400, 600 | $k_s \left[ \frac{W}{m \cdot K} \right]$ | 0.33 | 0.33 | (±0.02) |
| | $D_{Ts} \left[ \times 10^{-6} \frac{m^2}{s} \right]$ | 0.313 | 0.313 | (±0.019) |
| 200 | $k_s \left[ \frac{W}{m \cdot K} \right]$ | 0.48 | 0.53 | (±0.02) |
| | $D_{Ts} \left[ \times 10^{-6} \frac{m^2}{s} \right]$ | 0.265 | 0.313 | (±0.019) |

Apart from the evident conclusion that rise in crystallinity demonstrated the tendency of $D_{Ti}, k_s$ to increase, one could also say that the decrease in thickness facilitates the process of inverse solving, but also calls for caution when interpreting the dependence upon crystallinity due to the appearance of surface effects.

Another thing worth noticing is the significance of normalization, which was absent in this case (due to the influence of crystallinity) and which is proven to be very important for inverse solving of PA problems.

Finally, the relations among $D_{Ti}, k_s$ and crystallinity amplify the significance of future fundamental heat transfer investigations.

## 4. Conclusions

The subject of this work is the development of the techniques aiming at solving the inverse problem in photoacoustics. Its mid-term goal is the increase in the number of material properties which can be characterized by PA measurements with a satisfactory level of accuracy, while its long-term goal is the improvement of the methods of PA imaging of different materials, from macromolecule nanostructures and nanoelectronics or nanophotonic devices, to biological tissues.

Within the analysis of the TMS model, the influence of optical, thermal, and elastic properties of the sample were investigated. This analysis revealed that some of the processes, characterized by certain sample properties, exert their dominance only in limited modulation frequency ranges, which are shown to be dependent upon the choice of the sample material and its thickness. In the rest of the range their influence can be neglected, so the TMS model is divided into parts, each corresponding to the appropriate modulation frequency range.

The main conclusions of this progress report are gathered in the form of a bulleted list:

- Generalized model of PA response as the consequence of finite heat propagation velocity was considered and its manifestations—thermal resonances—were described, with potential application in the determination of heat propagation velocity by making use of the location of the first peak;
- The influence of multiple optical reflections on PA response was considered for a specific class of soft matter materials and its potential application, as well as implications regarding fundamental heat transfer were pointed out;
- Minimum volume PA cell was successfully modeled as Helmholtz resonator and innovative applications of PA methods were potentiated;
- Simultaneous use of amplitude and phase measurements was proven to enable the estimation of thermal diffusivity, while difficulties in assessing the ratio of linear expansion coefficient and heat conductivity coefficient pointed out the necessity for the improvement of TMS modeling;
- The application of a neural network on the numerical experiment exposed the necessity for the reconsideration of the thermal piston model in materials with low levels of arrangement (macromolecules, tissue, soft matter);
- The application of self-consistent procedures on the experiment demonstrated the dependence of thermal properties upon thickness and crystallinity.

**Author Contributions:** Experimental investigation, software modelling and writing—original draft preparation, M.N.; simulations and modelling, M.P.; conceptualization, methodology, formal analysis and writing—review and editing, S.G.

**Funding:** This research received no external funding.

**Acknowledgments:** We acknowledge the support of the Ministry of Science of the Republic of Serbia throughout all the explorations done as part of the III45005 project.

**Conflicts of Interest:** The authors declare no conflict of interest.

## Abbreviations

The following abbreviations are used in this manuscript:

| | |
|---|---|
| TMS | theoretically/mathematically simulated (TMS) models |
| PA | photoacoustic/photoacoustics |
| EM | electromagnetic |
| 1D | one-dimensional |
| HDPE | High-Density Polyethylene |
| WAXD | wide angle X-ray diffraction |
| DSC | diffraction scanning calorimetry |

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
