# Peer review of "Developing the Techniques for Solving the Inverse Problem in Photoacoustics"

_atoms, doi:10.3390/atoms7010024_

Round 1

Reviewer 1 Report

This communication is very hard to judge accurately as it is not presented in a clear and readable style. Despite reading it several times over a two week period I do not feel the paper, as currently written, allows me or other readers to determine the essential points the authors wish to make

I offer the following as a method for revising the text. Start by introducing the experiments to be analysed and what such an analysis may demonstrate as to effectiveness of previous analysis techniques. Then discuss the new analysis methods and tools with parameters that can be explored. Conclude with new findings and how the methods provide new insights into parameters of the samples.

I would also suggest the authors find someone more experienced in written English to help structure and write the paper. 

Author Response

We would like to express our gratitude to the Reviewer for having carefully read our work; we adopted all the suggestions and changed the paper accordingly. The list of alterations is given below:

1)      We have done extensive check-up of English language;

2)      We have improved  Introduction and added a few novel references;

3)      We improved the design of the paper in a manner that sections 2 and 3 consist of our results and their discussion:
Section 2 presents the implemented generalization of TMS models and discussion about dominant influences in our measurement frequency range;
Section 3 gives the description of two techniques for inverse solving and one example of the application of self-consistent procedure on experimental results, as well as the discussion about possibilities of application of the suggested techniques.

Section 4 consists of a brief description of our contributions, aims and the list of most important conclusions. We sincerely hope that our results are now more clearly presented and, consequently, that the conclusions can be more easily related to them.

Reviewer 2 Report

Interesting and useful work.

Several remarks – recommendations….

In Abstract it is incomprehensible phrase “…solving the exponential problems of mathematical physics…”

What does mean termal relaxation time in Eq. (3b)? I have not met in literature, devoted to hyperbolic equation of heat transfer, explanations of this parameter through other physical quantities. This is all the more important, since in publications [6],

Yu. Gurevich, G. Logvinov, L. Nino de Rivera, O. Titov, Nonstationary Temperature Distribution Caused by Bulk Absorption of Laser Pulse, Review of Scientific Instruments, 2003, Vol.74, p. 441-443,

and

Yuri Gurevich, Georgiy Logvinov, Igor Lashkevich, Effective Thermal Conductivity: Application to Photothermal Experiments for the Case of Bulk Light Absorption, Phys. Stat. Sol (b), 2004, Vol. 241, p. 1286-1298

this term is used in a different sense.

How were obtained Eqs (4)?

It would be interesting to compare the results of section 2.1.2. with works

Yuri Gurevich, Georgiy Logvinov, Igor Lashkevich, Effective Thermal Conductivity: Application to Photothermal Experiments for the Case of Bulk Light Absorption, Phys. Stat. Sol (b), 2004, Vol. 241, p. 1286-1298

and

G. N. Logvinov, Yu. G. Gurevich, I. M. Lashkevich, Surface Heat Capacity and Surface Heat Impedance: An Application to Theory of Thermal Waves, Jpn. J. Appl.Phys, 2003, Vol. 42, Part I, p. 4448-4452.

In Conclusions  the term "thermal resonance" appears (?).What it means?

(Compare with

Yu. G. Gurevich, G. N. Logvinov, N. Munoz Aguirre, L. Martinez Perez, “Resonance" Phenomena in Thermal Diffusion Processes in Two-Layer Structures, Applied Physics Letters, 2002, Vol. 80, p. 2898-2900.

Author Response

On behalf of the group of authors, I would like to express our gratitude to the Reviewer for having carefully read our work.

As the reviewer suggested, we have explained the expression “thermal relaxation time” and have introduced the proposed references. We agree with him that it would be interesting and useful to introduce a detailed comparison of our results with the ones present in the mentioned references, however, the ever-present and unfortunate limitation of the space in publications such as this one implies that we do it only in brief.

Of course, in our future publications we plan to include much detailed presentation of the comparisons and the conclusions which will be derived.

Round 2

Reviewer 1 Report

The authors have made signifcant changes to the text that have improved the clarity of the text

Some minor changes are still required, in several figures the text is not in English (Serbian?) which is not understandable to the vast majority of the readers. These should be corrected.

Author Response

Dear Reviewer,

On behalf of the group of authors signing this paper, I would like thank You for these final remarks: they are obviously the result of our omission, since an earlier version of this paper held all the Figures in English (it is my assumption that, at some point of the revision process, I did the corrections in the wrong file, which was subsequently uploaded as the corrected version of the paper). Thus, the appropriate alterations were done in the Figures: (2), (6), (9), (10) and (11).

Also, a small part of the text was added at the end of the paper, before References (according to the instructions of the official template), regarding funding, conflicts of interest and abbreviations – if considered unnecessary, this can be deleted in order to save print space. This portion of text is given in bright red.

With best regards, sincerely Yours,

Slobodanka Galovic

Marica Popovic,

Mioljub Nesic